# Structure of mammalian Mediator complex reveals Tail module architecture and interaction with a conserved core

Haiyan Zhao [1,5], Natalie Young[1,5], Jens Kalchschmidt[2,5], Jenna Lieberman[2], Laila El Khattabi[3], Rafael Casellas[2,4] & Francisco J. Asturias[1✉]

The Mediator complex plays an essential and multi-faceted role in regulation of RNA polymerase II transcription in all eukaryotes. Structural analysis of yeast Mediator has provided an understanding of the conserved core of the complex and its interaction with RNA polymerase II but failed to reveal the structure of the Tail module that contains most subunits targeted by activators and repressors. Here we present a molecular model of mammalian (*Mus musculus*) Mediator, derived from a 4.0 Å resolution cryo-EM map of the complex. The mammalian Mediator structure reveals that the previously unresolved Tail module, which includes a number of metazoan specific subunits, interacts extensively with core Mediator and has the potential to influence its conformation and interactions.

[1] Department of Biochemistry and Molecular Genetics, University of Colorado Anschutz Medical School, Aurora, CO, USA. [2] Lymphocyte Nuclear Biology, NIAMS, NIH, Bethesda, MD, USA. [3] Institut Cochin Laboratoire de Cytogénétique Constitutionnelle Pré et Post Natale, Paris, France. [4] Center for Cancer Research, NCI, NIH, Bethesda, MD, USA. [5] These authors contributed equally: Haiyan Zhao, Natalie Young, Jens Kalchschmidt. ✉email: Francisco. Asturias@CUAnschutz.edu

Mediator, a large multi-protein complex first identified in budding yeast as an agent required to convey regulatory information to the basal transcription machinery[1–3], plays an essential and multi-faceted role in regulation of RNA polymerase II (RNAPII) transcription in all eukaryotes[4,5]. A core Mediator comprising Head (7 subunits) and Middle (8–9 subunits, depending on the specific organism) modules, and a central MED14 subunit, is conserved across eukaryotes. The Cdk8 kinase, the only Mediator subunit with well-defined catalytic activity[6], associates with three other subunits to form a dissociable kinase module (CKM). Finally, a Tail module (2–9 subunits, again depending on the specific organism) is considerably more variable and contains many subunits targeted by activators and repressors. Perhaps reflecting a need for more nuanced regulation, five subunits (MED23, MED25, MED26, MED28, and MED30) and several paralog components of the CKM are present only in higher eukaryotes. The contrast between the intricate and essential role of Mediator in transcription and its minimal catalytic activity highlights the significance of understanding Mediator's structure and interactions. Early studies of yeast and mammalian Mediator revealed a well-defined but dynamic structure[7,8], and direct interactions with RNAPII that seemed congruent with Mediator's role in transcription initiation. Most structural studies have been focused on yeast Mediators, whose structure and interaction with RNAPII are now reasonably well understood[9–11]. However, information about mammalian-specific aspects of Mediator structure and function is essential to understand metazoan transcription regulation.

A limited structural understanding of mammalian Mediator (mMED) represents an obstacle to interpreting the large body of functional and biochemical observations about the complex. Structures of only a very small number of isolated mammalian Mediator subunits and domains have been reported to date, and a detailed and holistic understanding of the complex has remained elusive[12,13]. We recently determined an intermediate (6 Å) resolution cryo-EM structure of mMED that provided insight into its molecular organization, explored the function of Tail subunits, and clarified the role of Mediator in enhancer–promoter contacts[14]. In this report, we present a molecular model of mMED derived from a 4.0 Å resolution cryo-EM map of the complex. The model shows how metazoan-specific subunits have been integrated and add to the complexity of mMED organization, influencing its conformation and interactions.

## Results

**A molecular model of mammalian Mediator.** Endogenous tagging of Mediator subunits in mouse CH12 cells[14] allowed us to use an immunoaffinity purification approach to obtain Mediator fractions suitable for cryo-EM analysis of the complex (Supplementary Fig. 1 and Supplementary Table 1). We used state-of-the-art cryo-EM analysis to build on our published low-resolution (~6 Å) structural analysis of the mammalian Mediator complex[14] and were able to obtain a cryo-EM map with an overall resolution of 4.0 Å (Fig. 1a, Supplementary Figs. 2 and 3, and Supplementary Table 2). Portions of the Head and Tail modules and MED14 reached a maximum resolution of ~3.5 Å (Fig. 1b). The Middle's hook and the Head's neck, known to interact with the RNA polymerase II (RNAPII) carboxy-terminal domain (CTD)[9,15] showed the highest conformational variability. Secondary structure elements were clearly resolved throughout the cryo-EM map and densities for bulky amino acid side chains were apparent throughout MED14, the lower portion of the Head module, and the Tail module. Following on our previous naming convention, we assigned all non-core/non-CKM subunits to the Tail module (Supplementary Table 3).

The 4.0 Å resolution mMED cryo-EM map and information from published structures of yeast Mediator[9–11] allowed us to determine the structures of 25 out of 26 non-kinase module mMED subunits, including 16 core subunit, 7 Tail subunits (MED15, MED16, MED24, MED27, MED28, MED29, and MED30) whose structure and precise location and interactions were previously unknown, and 2 Tail subunits (MED23 and MED25) whose structures were partially (MED25[13]) or completely (MED23[12]) known, but only as individual proteins outside the context of the entire Mediator. Focused refinement of densities at either end of the Middle module provided additional insight into the structure of the hook (MED14 N-terminus, MED10, and MED19) and the N-terminal portion of MED1. Combining all of this information resulted in a molecular model including 24 mMED subunits, and a partial MED1 model (Fig. 1c, Supplementary Movie 1 and Supplementary Table 3). This left MED26 as the only mMED subunit for which only general interaction information, but no detailed structure is available. Information about the structure of the four subunits in the Cdk8 kinase module (CKM) has come from studies in yeast[16] and interaction of the CKM with mMED has been characterized by EM[16,17].

Several features of the mMED model are worth noting. First, the structure and organization of core mMED subunits is remarkably conserved from yeast to mammals, but yeast and mammalian core Mediators differ considerably in conformation. Second, a centrally positioned mammalian MED14 enables inter-module interactions, as observed in yeast, but mammalian MED14 has a much larger C-terminal portion that is heavily involved in Tail interactions. Third, core subunit domains (MED14 N-terminus, MED17 N-terminus, MED6 C-terminus) that cross-module boundaries to make inter-module connections that link the Middle's knob and Head's neck domains are conserved between yeast and mammals. Fourth, the structures of MED14 and each module show particular characteristics. The Head shows a highly integrated structure, with considerable intertwining of conserved subunits with metazoan-specific ones, and extensive interfaces with MED14 and the Tail. The Middle shows an extended structure, with component subunits that, with the exception of MED1, are almost exclusively helical, resulting in overall rigidity and the potential for conveying long-range structural rearrangements. The Middle's interaction with other modules is limited to focused contacts. The Tail is divided into upper and lower portions[14]. Subunits in the lower Tail are large and mostly self-contained. In contrast, smaller subunits in the upper Tail have extended structures and large interfaces amongst themselves and with MED14 and the Head.

**Structure and conformation of the mammalian Head module.** The structure and subunit organization of the mammalian Head module are very similar to those of the yeast Head, with MED17 acting as the central scaffold for Head assembly (Fig. 2a). Mammalian MED17 is larger than its yeast counterpart due to the presence of an insertion (aa 384–419) and a ~110 residue C-terminal extension (aa 540–650) (Fig. 2b). Both of these mammalian-specific MED17 domains are involved in interactions with other subunits, with the insertion forming an extended interface with the Head's MED18–MED20 subcomplex (Fig. 2c) and the longer C-terminus adopting an extended structure involved in interaction with MED14 and the Tail. While some Head subunits like MED18–MED20 are nearly identical in structure and conformation between yeast and mouse (Fig. 2d, top), others like MED8 adopt the same structure but show a different conformation (Fig. 2d, bottom). Mammalian Head subunits MED11 and MED22 also differ from their yeast

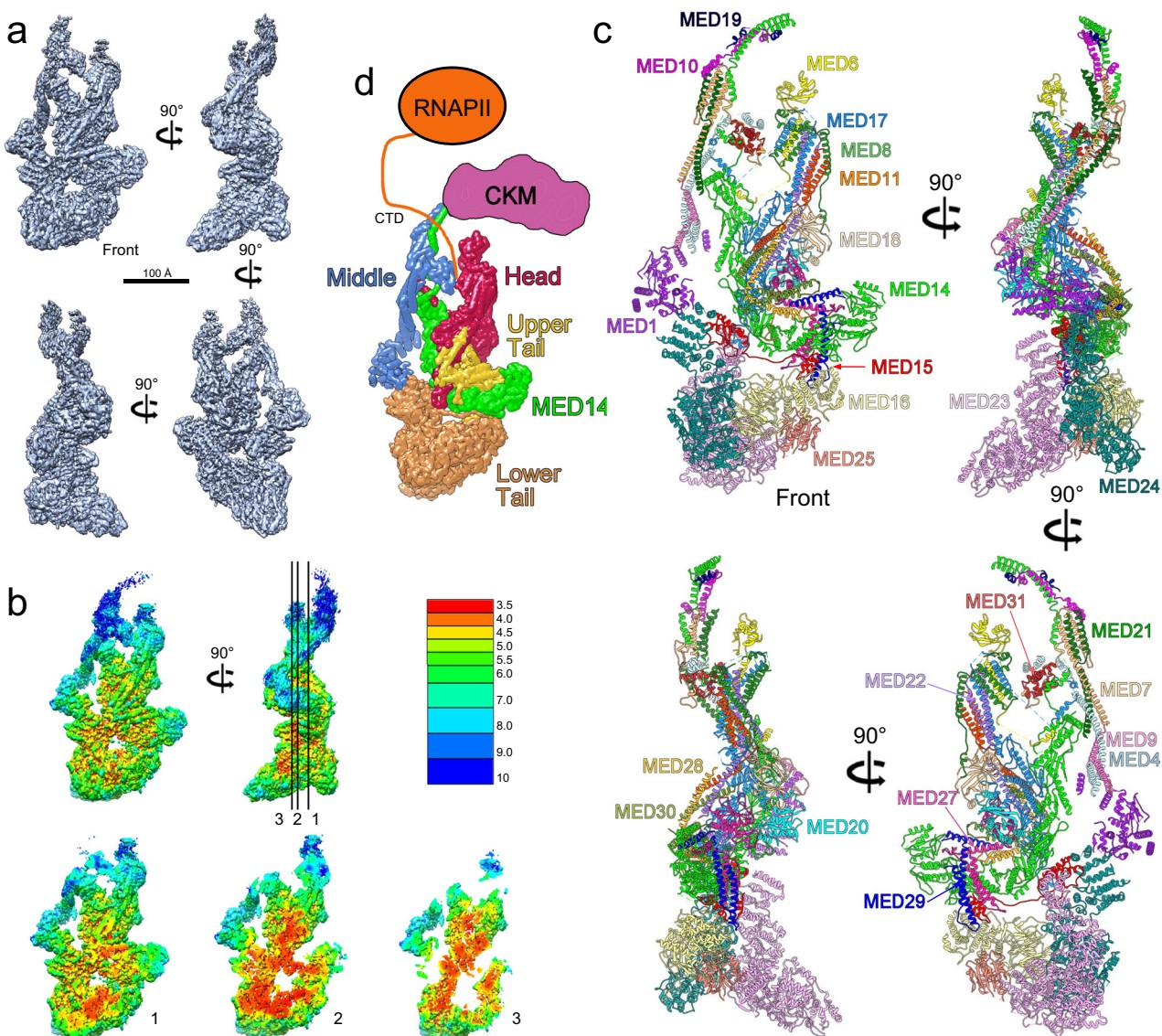

**Fig. 1 4.0 Å resolution cryo-EM map and molecular model of mammalian Mediator. a** mMED cryo-EM map at 4.0 Å resolution. **b** Local resolution in the mMED map points to high mobility of the Middle module and domains that form the CTD-binding gap. Slicing through the map highlights comparatively high (~3.5 Å) resolution throughout the core of the complex. **c** Mammalian Mediator molecular model. **d** Schematic representation of mMED's modular organization and its interaction with RNA polymerase II (RNAPII, through the polymerase CTD) and the Cdk8 kinase module (CKM).

counterparts in conformation and by having longer, well-ordered C-terminal helices (Fig. 2e). Reflecting the similarity and differences between corresponding subunits, the yeast and mammalian Head modules have the same overall structure but adopt a different conformation. This difference in conformation results from a change in the relative positions of the top (neck) and bottom (jaws) portions of the Head (Fig. 2f). explained by flexing at the connection between the two, which is solely composed of extended loop domains that allow all of the subunits that cross the interface to flex (Fig. 2f, inset).

**Structure and conformation of the mammalian Middle module and MED14.** The mammalian Middle is similar to the yeast Middle (Fig. 3a), with a notable difference being that the N-terminal portion of MED1 is much better ordered (likely due to strong contacts with the Tail) and partially resolved in the mMED cryo-EM map. The central portion of the mammalian Middle can be divided into top and bottom sections formed by long helical bundles formed, as seen in yeast, by MED7–MED21 (top) and

MED4–MED9 (bottom) (Fig. 3b). MED31, the Mediator subunit showing the highest sequence conservation across eukaryotes, also shows high structural similarity between yeast and mammals (Fig. 3b, bottom panel). As observed for the Head, individual corresponding subunits are very conserved structurally, yet the overall conformation of the yeast and mammalian Middle modules differ. This is again due to flexing, in this case at the MED7–MED21/MED4–MED9 interface (Fig. 3c, see inset for details). Some flexibility of the Middle module structure was suggested by X-ray analysis of *S. pombe* Mediator[11].

The distal portions of the Middle module are only partially ordered (6–10 Å) and were further analyzed through focused refinement. As anticipated based on localization by subunit deletion[10,14], MED1 forms the lower end of the Middle module and in mMED has extended contacts with MED24 in the Tail (Supplementary Fig. 4a). The very N-terminus of MED1 wraps around MED4–MED9 and the subunit is composed of alternating α-helical and β-sheet domains (Supplementary Figs. 4b and 5). The tertiary organization of MED1 is reminiscent of MED14 (see

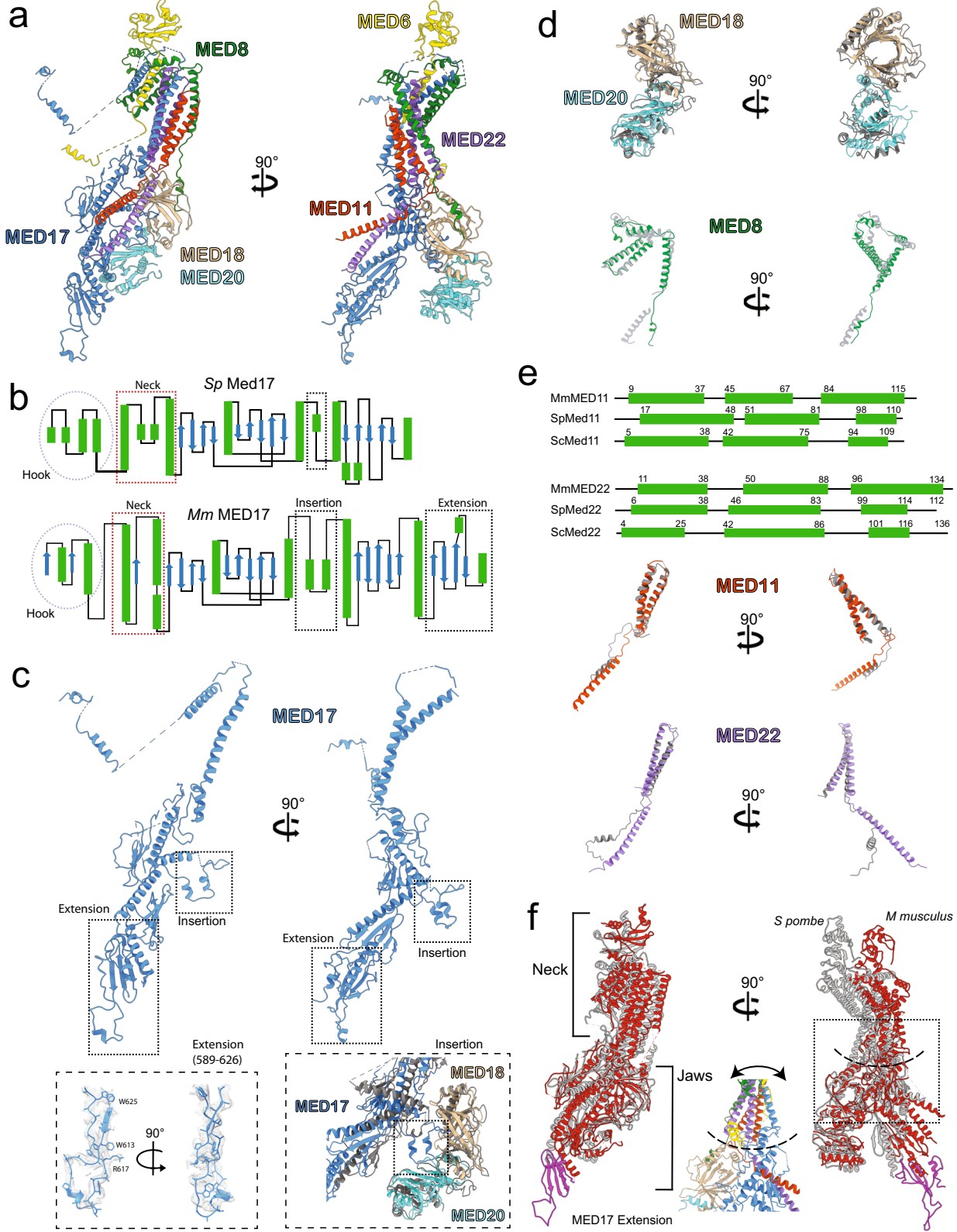

below) and might result in flexibility that would explain why the MED1 portion of the cryo-EM map is poorly resolved (~6–8 Å) in comparison to other regions. Nonetheless, the partial MED1 model is consistent with recently reported cross-linking mass spectrometry data[18] and provides insight into the structure of this important nuclear receptor target in the context of the larger Mediator structure. Focused refinement of

the knob/hook resulted in comparatively improved density for the domains (Supplementary Fig. 6), but they remained poorly resolved and molecular models for the corresponding subunits are informed by consideration of secondary structure similarity to their yeast counterparts and the published X-ray structure of the yeast Mediator knob and hook[11] (Supplementary Fig. 7).

**Fig. 2 Structure of the mammalian Head module. a** Overall mammalian Head module structure. **b** Comparing the secondary structure of yeast (*S. pombe*) and mammalian (*M musculus*) MED17 proteins highlights their similarity and reveals the presence of an insertion and a C-terminal extension in the mammalian protein. **c** Like its yeast counterpart, MED17 functions as the central scaffold subunit for the Head module and shows a similar overall structure. A mammalian-specific insertion interacts with MED18/MED20 (bottom, right), while a C-terminal extension, also mammalian-specific, adopts an extended, well-ordered structure (bottom, left). **d** The structure of mammalian MED18–MED20 is remarkably similar to that of their yeast counterparts (yeast in gray). The MED8 structure is also conserved, but the conformation of the yeast and mammalian proteins differ (yeast in gray). **e** The structures of mammalian MED11 and MED22 are mostly conserved (yeast in gray), but mammalian MED11 and MED22 have longer, well-ordered C-terminal α-helices that are involved in interaction with metazoan-specific upper Tail subunits. **f** Comparing the mammalian (red) and yeast (gray) Head modules shows a change in overall conformation due to flexing at the neck–jaws interface, which is formed by flexible loops in all subunits that cross the interface (inset).

The theme of subunit structure conservation continues with MED14. Sequence analysis predicted similarities between corresponding N-terminal portions of mammalian and yeast MED14[14] and the two are in fact remarkably similar in both secondary structure and tertiary organization (Fig. 3d). The alternating pattern of α-helical and β-sheet domains seen in yeast Med14 is also present in mammalian MED14 and, in fact, that pattern continues into the much larger (mouse MED14 1454 aa, *S. cerevisiae* MED14 1082 aa, *S pombe* MED14 879 aa) C-terminal portion of the mammalian MED14 (Fig. 3d, highlighted by red rectangle), which is heavily involved in interactions with the large (~790 kDa) mammalian Tail module. Modeling of the extended and intricate structure of mammalian MED14 was made possible by the presence of considerable detail in the cryo-EM map (Fig. 3d, inset). Mammalian MED14 extends nearly all the way across mMED, with its N-terminus forming part of the hook at the top of the Middle module and its C-terminus located over >350 Å away at the interface between the upper and lower portions of the Tail (Fig. 3e).

Although sequence homology between yeast and mammalian Mediator subunits is generally low (<25% on average), the structures of individual subunits and modules are highly conserved. Flexing of the mammalian Head and Middle modules at their hinge regions results in a remarkable correspondence between the structures of the yeast (*S pombe*) and mammalian (*M musculus*) core Mediators.

**Structure of the mammalian Tail module**. Information about the structure of the Tail has been limited to approximate localization of Tail subunits in the context of low (16 Å *S cerevisiae* Mediator[10]) or intermediate (6 Å *M musculus* Mediator[14]) cryo-EM maps, a proposed *S cerevisiae* Tail molecular organization map based on integrative modeling[19], tentative localization of part of the yeast Med27 C-terminus (in a 4 Å *S pombe* Mediator cryo-EM map[9]), and a recently published X-ray structure of human MED23[12]. Our mMED molecular model (Fig. 1c) provides a detailed view of the entire Tail.

The upper Tail forms an extended connection between core Mediator (specifically the Head and C-terminal portion of MED14) and the lower Tail (Fig. 4a, left). The four subunits forming the upper Tail (MED27–30) are similar in size and structure. MED28–29–30 are all 180–200 aa long, and so is MED27 if a globular C-terminal domain (~100aa) is considered separately. Despite their structural similarity, upper Tail subunits could be distinguished from one another and identified based on differences in the length of specific helices and loops and the presence of bulky side-chain densities (Supplementary Figs. 8 and 9). The upper Tail subunits are organized in pairs (MED27–MED29 and MED28–MED30) to form a double bracket-like structure (Fig. 4a, right). As observed in *S pombe*[9], the MED27 C-terminal globular domain sits between the jaws of the Head module, adding to the complexity of the Tail-core interface. In all four proteins, an N-terminal 2-helix coiled coil is followed by a short loop and a third α-helix arranged roughly

perpendicular to the N-terminal coiled coil (Fig. 4b). The N-terminal coiled coils of MED27–29 contact the C-terminal portion of MED16 in the lower Tail and then wrap around the backside of the MED14 C-terminus towards the Mediator core. The α-helices that follow the N-terminal coiled coil in both MED27 and MED29 interdigitate with the MED28–30 coiled coils to form a 6-helix bundle. Finally, the C-terminal helices in MED28–30 move toward the Head jaws where they form a 4-helix bundle with C-terminal helices from MED11 and MED22. The helical bundles in the upper Tail are stabilized by extensive hydrophobic interactions (Fig. 4c).

As observed in *S cerevisiae* Mediator[10], the mammalian lower Tail is organized around MED16, which establishes contacts with all other lower Tail subunits (Fig. 5a). MED16 has a bipartite structure (Fig. 5b), with its N-terminal portion forming a large β-propeller, as predicted[19] (Fig. 5b, bottom right), and its C-terminal portion being entirely α-helical (Fig. 5b, left). The massive distal end of the lower Tail, abutting the MED16 β-propeller, is formed by MED23 and MED24, which have roughly homologous structure and organization (Fig. 5c). Nestled between MED23, MED24, and MED16 is the folded N-terminal von Willebrand domain of MED25[20] (aa 15–216, Fig. 5a, b). No density corresponding to the C-terminal portion of MED25 (accounting for over two-thirds of the protein) was detected in the mMED cryo-EM map, which is explained by the presence of a disordered loop (aa ~198–392) connecting the folded N-terminal and C-terminal domains (Fig. 5d, bottom).

The last subunit in the mMED Tail, MED15, shows perhaps the most peculiar structure and arrangement. With the exception of a small N-terminal folded domain, the first ~530 MED15 residues are expected to be disordered (Fig. 5e, top) and were not detected in the mMED cryo-EM map. However, the C-terminal portion of MED15 was well resolved (Fig. 5e, middle). The first portion of MED15 visible in the cryo-EM map (residues 620–652) forms two short, roughly anti-parallel alpha helices (Fig. 5e, middle and bottom left). These helices are followed by a long loop (residues 654–674) that travels across the lower Tail module along the MED14 C-terminus to connect to a folded domain at the very C-terminus of MED15 (residues 678–787, Fig. 5e, middle and bottom right). This unusual and very extended organization of MED15 was confirmed by maltose-binding protein (MBP) labeling of the MED15 C-terminus (Fig. 5f). The folded domain at the MED15 C-terminus sits between the N-terminal portions of MED23 and MED24 (Supplementary Fig. 10a). Other subunit interactions between lower Tail subunits happen through a combination of charged, hydrophobic and cationic-π interactions (Supplementary Fig. 10b). Interestingly, peripheral subunits in the lower Tail are rather self-contained. For example, both MED23 and MED25 can be absent without compromising Tail integrity or its interaction with core mMED[14]. This could reflect relatively recent incorporation of these subunits into Mediator.

**Disease-associated mMED mutations**. The mMED model provides information about the location of known disease-associated

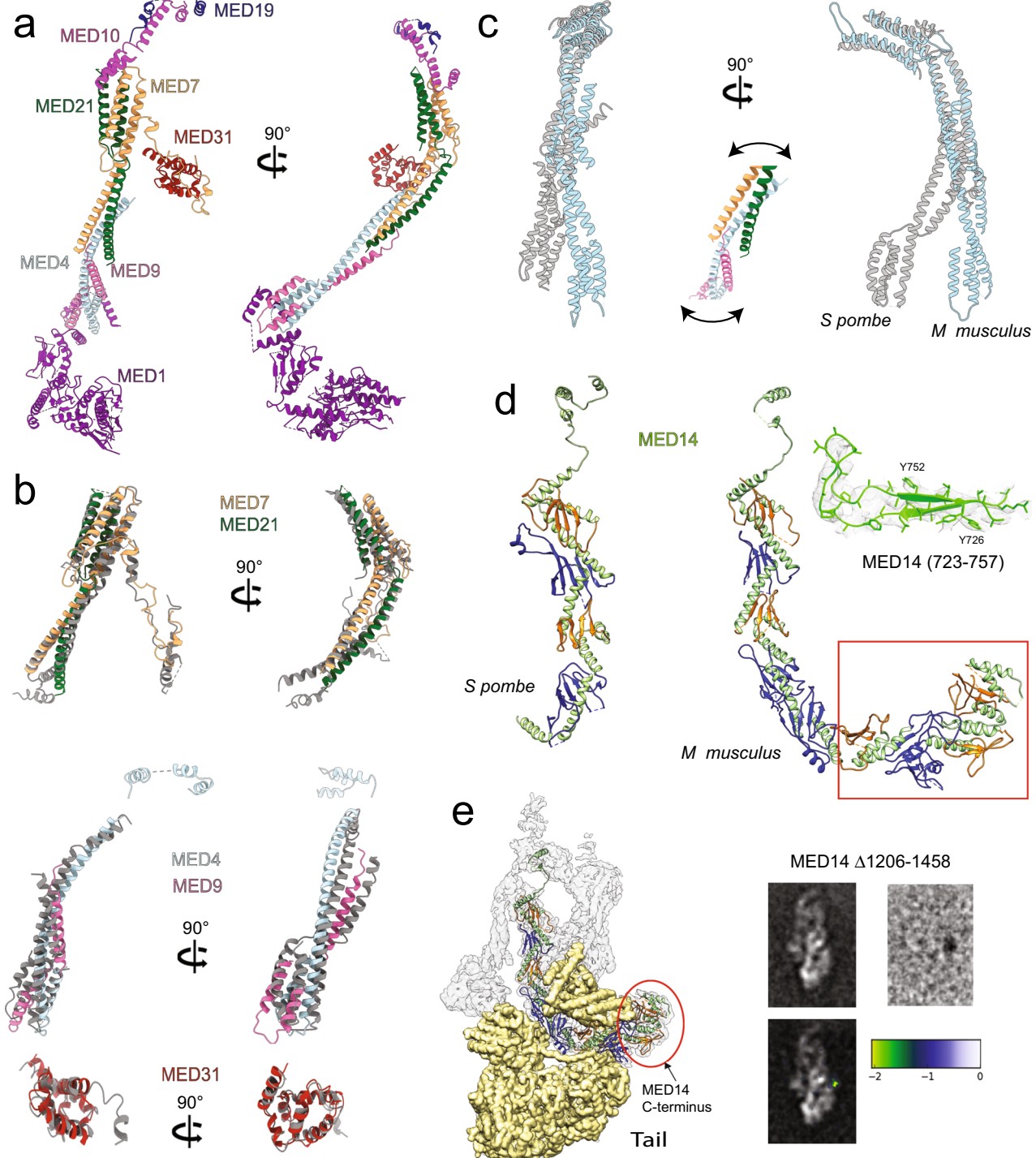

**Fig. 3 Structure of the mammalian Middle module and MED14. a** Overall mammalian Middle module structure, including a partial MED1 model. **b** The structure of MED7–MED21 (top), MED4–MED9 (middle) and MED31 (bottom) are mostly conserved between yeast and mammalian Mediators (yeast subunits in gray). **c** The conformation of the mammalian Middle (light blue) differs considerably from that of the yeast Middle (gray) due to flexing at the MED7–MED21/MED4–MED9 interface (see inset for details). **d** The structure of mammalian MED14's N-terminal portion structure is remarkably similar to the corresponding portion of yeast MED14, but the mammalian subunit is considerably larger due to an extended C-terminus (highlighted by red rectangle). Elucidation of the MED14 structure was facilitated by considerable detail in the cryo-EM map (inset). **e** The mammalian MED14 C-terminus interacts extensively with the Tail module (left). The position of MED14's extreme C-terminal portion (highlighted by red ellipse on the left) was confirmed by difference mapping of a MED14 Δ1206–1458 mMED mutant. A MED14 Δ1206–1458 mMED class average and the difference map obtained after subtracting an aligned mMED class average are shown in the top right. Difference map densities colored by standard deviation (as indicated in the color scale) and superimposed on a mMED class average are shown in the bottom right.

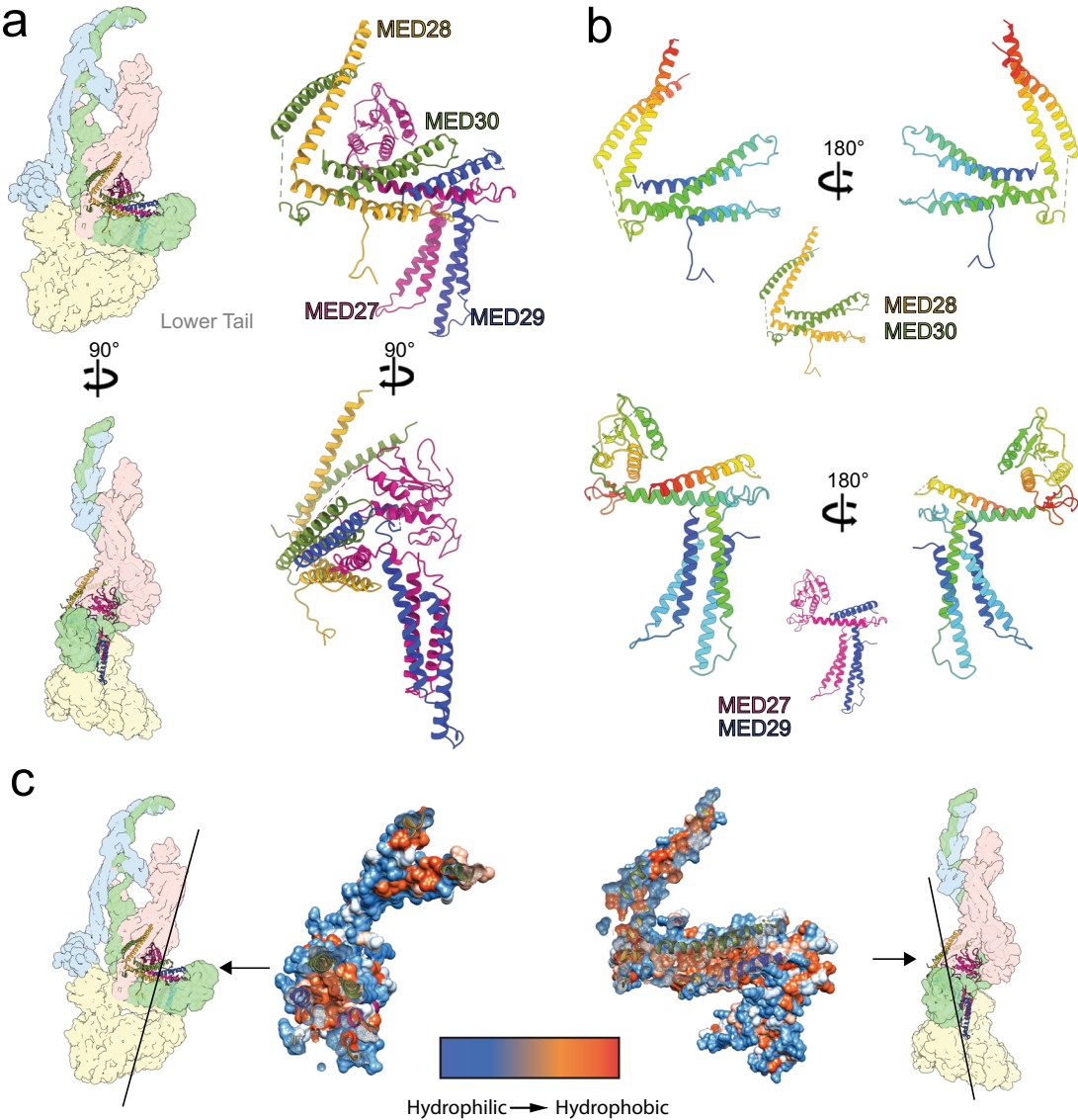

**Fig. 4 Structure of the mammalian upper Tail. a** Overall upper Tail structure and its position within the mMED structure (Head light red, MED14 light green, Middle light blue, and lower Tail light yellow). **b** The structure of all four upper Tail subunits (MED27–30) is remarkably similar, with the only deviation amongst them being the presence of a globular domain at the MED27 C-terminus. Subunit color changes from blue (N-terminus) to red (C-terminus). **c** Formation and stability of helical bundles in the upper Tail is driven by hydrophobic interactions between subunits. This is shown by slicing the upper Tail structure as indicated by the black lines.

mutations in Mediator subunits (Supplementary Fig. 11), which can provide insight into their effect. For example, the human L371P MED17 mutation is associated with postnatal onset microcephaly[21]. The corresponding mouse mutation, MED17 L369P, would sit near the end of a helix that interfaces with the helical bundle formed by MED11–MED22 (Head) and MED28–MED30 (upper Tail). The leucine to proline mutation would very likely alter the MED17 helix and could affect the conformational dynamics of the Head. Consistent with this hypothesis, the corresponding mutation in *Sc* Mediator (*Sc*MED17 M504P) did not destabilize the Head module, but impaired Mediator–RNAPII interaction[22]. Similarly, decreased MED25 association with Mediator resulting from a MED25 Y39C mutation in the folded N-terminal MED25 domain that is linked to a severe genetic syndrome[23] is explained by our observation that the MED25 N-terminal domain interacts directly with other subunits in the lower Tail.

**Tail-core interfaces in mammalian Mediator**. The mammalian Tail has an extensive and intricate interface with core Mediator, wrapping around the entire bottom of the core (Fig. 6a). The upper Tail has a convoluted interface with Head subunits, with the C-terminal α-helices of metazoan-specific subunits MED28 and MED30 forming a helical bundle with the extended C-terminal helices in MED11 and MED22 (Fig. 6b). Additional complexity in the upper Tail–Head interface comes from the globular C-terminal domain of MED27, which interacts with MED18–MED20 and is close to the MED17 insertion domain (Fig. 6c). The upper Tail also interacts with MED14, as MED27–MED29 wrap around it to reach down towards MED16 and interdigitate with the two short, antiparallel helices formed by MED15 residues 620–652, which end up positioned between the MED27 and MED29 coiled coils (Fig. 6d). This is consistent with well-established interaction of MED15 with MED27 (Med3) and MED29 (Med2)[10,19,24,25]. The well-ordered conformation of the

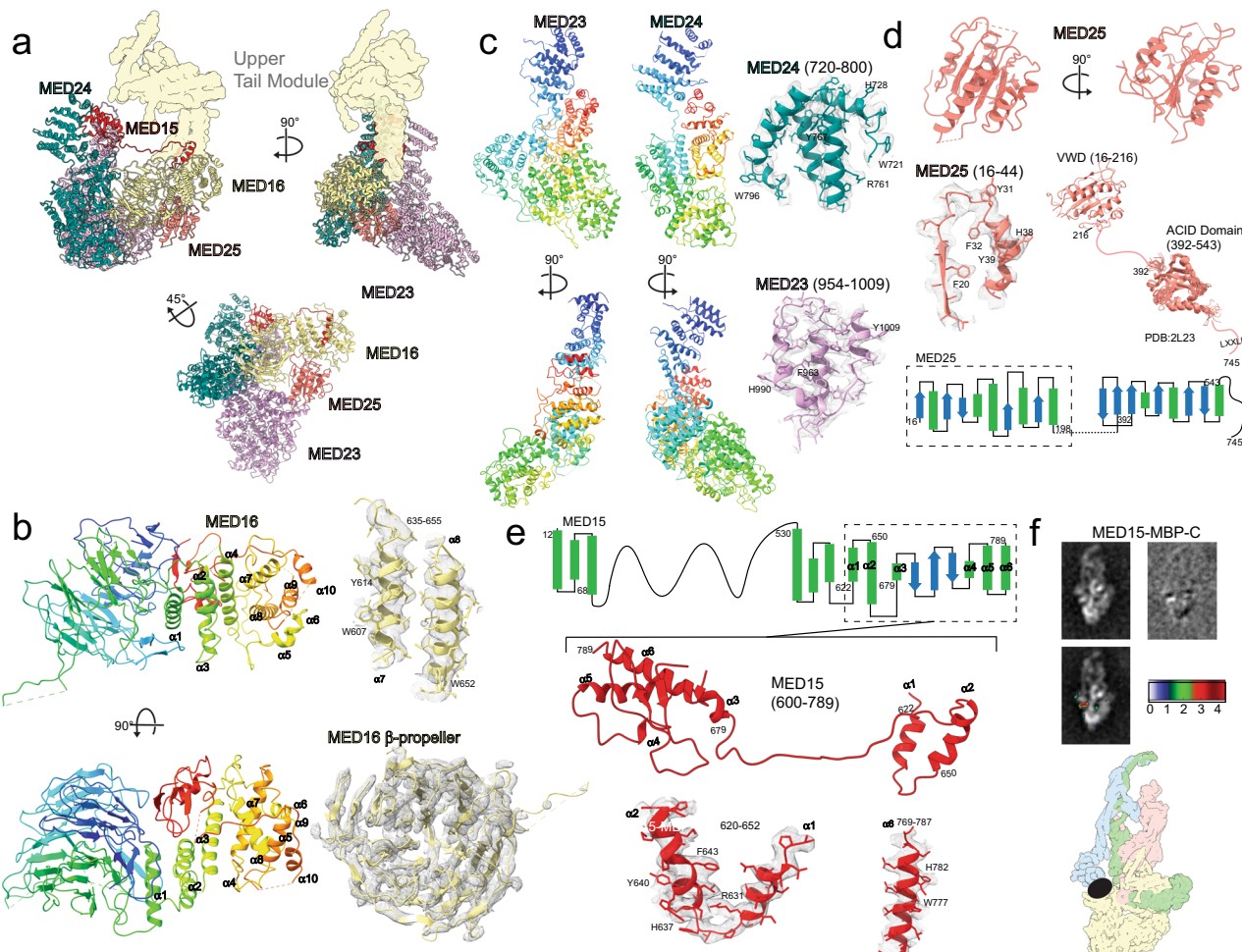

**Fig. 5 Structure of the mammalian lower Tail. a** Overall lower Tail structure (upper Tail in light yellow). **b** MED16 structure with β-propeller N-terminal and α-helical C-terminal domains colored from blue to red (top left) and map-to-model comparisons (top right), and details of the N-terminal β-propeller domain (bottom). **c** Models for MED23 and MED24 (left) show remarkable similarity in overall organization of the subunits (subunit color changes from blue (N-terminus) to red (C-terminus)) and correspondence between the models and the cryo-EM map (right). **d** Structure of the N-terminal von Willebrand MED25 domain (top) that interacts with other lower Tail subunits, and map-to-model comparison (middle left). The MED25 C-terminal activator interaction domain (ACID, PDB 2L23) is connected to the von Willebrand domain by a long flexible loop (middle left and bottom) and was not detected in the cryo-EM map. **e** Secondary structure diagram for MED15 (top) showing that its N-terminal portion is expected to be mostly disordered. The C-terminal portion evident in the mMED cryo-EM map (middle) is highlighted by the dashed rectangle and starts with two short anti-parallel α-helices (bottom, left) connected to a folded domain at the very C-terminus by an extended loop. **f** Localization of the MED15 C-terminus by MBP labeling and difference mapping. A MED15-MBP mMED class average (left) and the difference map obtained after subtraction of an aligned mMED class average from it (right) are shown in the top row. The middle row shows difference map densities colored by standard deviation (as indicated in the color scale) and superimposed on a mMED class average. The bottom row is a diagram showing the position of the MED15 C-terminus in the overall mMED structure.

extended MED15 loop is explained by extensive and specific interactions with MED14, with charged residues in both subunits aligned opposite to one another along the entire length of the loop (Fig. 6e, top right). In addition to extensive contacts with MED23 and MED24, the C-terminal folded MED15 domain also has an extended interface with MED14 (Fig. 6e, bottom right), and is contacted by the mammalian-specific C-terminal extension in MED17 (Fig. 6f). This very extended MED15 structure results in the subunit running all the way across the Tail from the MED23–24 N-terminal domains to the MED27–28–29–30 assembly on the opposite side, putting MED15 at the center of the core–Tail interface. Finally, crosslinking-mass spectrometry analysis of yeast Mediator pointed to interaction between MED1 and MED24 (yeast Med5)[19] and the mMED structure provides additional insight into the MED1–MED24 interaction. Although we cannot generate a detailed model of the folded N-terminal portion of MED1, interpretation of the MED1 focused refinement

map suggests that β-strands rich in hydrophobic residues formed by aa ~437–481 in the folded MED1 N-terminal domain interact with a hydrophobic patch on the surface of the MED24 N-terminus (Fig. 6g).

**Tail effect on core mMED conformation**. The mammalian Mediator structure shows high interconnectivity of component subunits, which stabilizes the conformation of most of the complex. Compared to yeast Mediator, the presence of metazoan-specific subunits that form the upper Tail, along with extensive subunit interactions across the entire mammalian Mediator, result in a more conformationally stable complex. This is evidenced by local resolution analysis of the mMED cryo-EM map (Fig. 1b), which points to a generally stable structure, with the notable exception of the Middle's knob/hook and Head's neck domains involved in interaction with RNAPII and the CKM

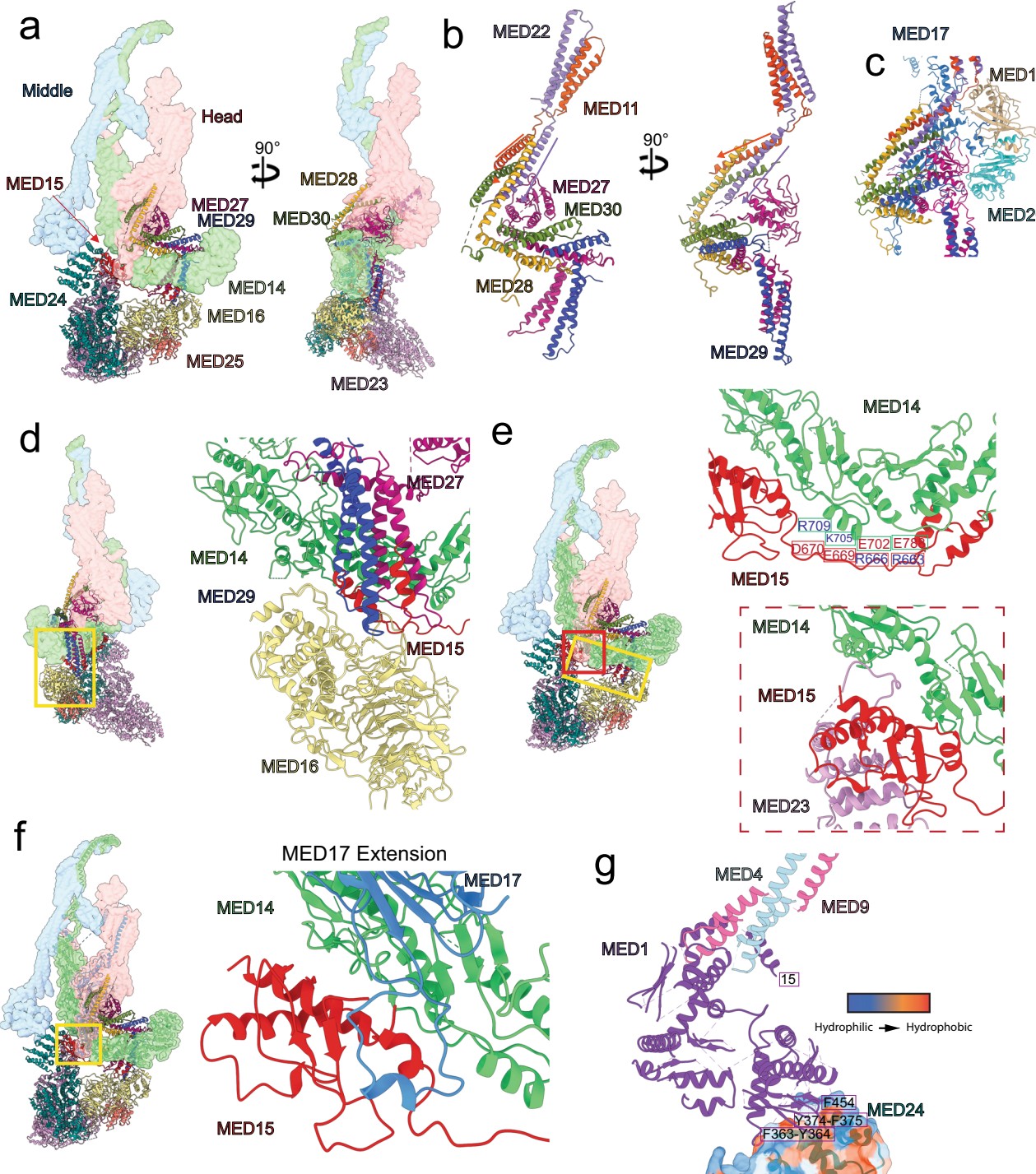

**Fig. 6 Tail–core interactions. a** Overall lower Tail structure and its interaction with core mMED (Head light red, MED14 light green, Middle light blue). **b** Interaction of MED28–MED30 in the upper Tail with MED11–MED22 in the Head. **c** Interaction of the upper Tail's MED27 C-terminus with MED18–MED20 and MED17 in the Head. **d** Interaction of the helical bundle formed by helices in MED27–29 (upper Tail) and MED15 (lower Tail) with MED14. **e** Charge interactions between an extended MED15 loop and the MED14 C-terminus (top), and interaction of a globular MED15 C-terminal domain (nestled between MED23 and MED24) with MED14 (bottom). **f** Details of interactions established by the MED15 C-terminal domain with MED14 and the MED17 C-terminal extension. **g** Interaction between MED1 N-terminal domain β-strands (aa ~437–481) rich in hydrophobic residues with a hydrophobic patch on the surface of the MED24 N-terminus.

(Fig. 1d). Interestingly, although individual yeast and mammalian core Mediator subunits have very similar structures, overall core conformation differs between the yeast and mammalian Mediators, primarily due to rearrangements in the Head and Middle modules facilitated by their intrinsic pliability.

For yeast Mediator there is strong evidence that changes in core conformation are necessary for formation of a Mediator-RNAPII holoenzyme[9]. We previously reported[14] that release of the Middle–Tail interaction by deletion of MED1 (the only direct interaction between the Middle and Tail modules is the contact

between Middle subunit MED1 and Tail subunit MED24) leaves the Tail unchanged but allows core mammalian Mediator to adopt a different conformation in which the CTD-binding gap between the knob and the neck (Fig. 1d) narrows. Importantly, this is accompanied by a considerable (~3-fold) increase in mMED interaction with RNAPII[14]. In mammalian Mediator, rigidity of the Tail structure and the nature of Tail–core interactions, both explained by the mMED molecular model, seem to enable the Tail to influence core mMED conformation. A rigid upper Tail that includes metazoan-specific subunits and displays a large interface with the Head and an extended MED14 C-terminus, keeps the lower Tail in a fixed position. In turn, attachment of the Middle to the Tail through the MED1–MED24 contact keeps the mammalian core Mediator in a specific conformation. Considering the effect of MED1 deletion on mMED core Mediator conformation and polymerase interaction suggests that a primary effect of the mammalian Mediator Tail could be to modulate Mediator function by biasing core Mediator conformation towards a state that limits interaction with RNAPII. Although we only obtained EM data for MED1-related effects, the mMED molecular model indicates that any change in Tail structure that decreased Tail rigidity or its interaction with the core, could potentially lead to changes in core conformation.

## Discussion

The structure of mMED reveals a remarkable conservation of core Mediator and suggests how tight integration of metazoan-specific subunits into the core could contribute to more nuanced Mediator-dependent regulation in higher eukaryotes. An important question is how changes in core Mediator conformation contribute to the overall Mediator mechanism. We previously reported results from EM analysis of mMED subunit deletion mutants showing that release of Middle–Tail interactions by deletion of MED1 (eliminating the contact between Middle subunit MED1 and Tail subunit MED24), or disruption of the Tail's integrity by deletion of Tail subunits, resulted in increased Mediator–RNAPII interaction[14]. In the case of MED1 deletion, we were able to show that the increase in RNAPII interaction might be explained by a change in the conformation of mMED. Whereas the structure of the Tail seemed to remain constant after MED1 deletion, core conformation changed, with the knob and the neck moving towards each other and narrowing the CTD-binding gap (Fig. 7a), which would presumably favor increased Mediator interaction with RNAPII. Increases in Mediator–RNAPII interaction were also seen upon deletion of various Tail subunits and subsequent changes at the large interface between the Tail (particularly metazoan specific-subunits, Fig. 7b) and core Mediator. These observations are consistent with the idea that changes in core conformation are important for the Mediator mechanism and suggest that targeting of MED1 or Tail subunits by transcription factors (TFs) could, at least in principle, affect Mediator interactions. However, until now, there is no conclusive structural evidence for Mediator conformational rearrangements triggered by TF interaction and further investigation is needed to address that possibility.

Interestingly, our model of MED1 organization (Supplementary Fig. 4a, b) suggests that the MED1 LXXLL motifs targeted by nuclear receptors[26] are outside the folded portion of the subunit, with the LXXLL motif closest to the folded N-terminal portion of MED1 ~80 aa past the last MED1 density visible in the mMED cryo-EM map. Similarly, the partially folded N-terminal MED15 domain targeted by acidic activators[27,28] is separated from the ordered C-terminal portion of MED15 by a >300aa intrinsically disordered region (IDR), and the well-folded C-terminal ACID domain of MED25 targeted by VP16, ERM, other members of the

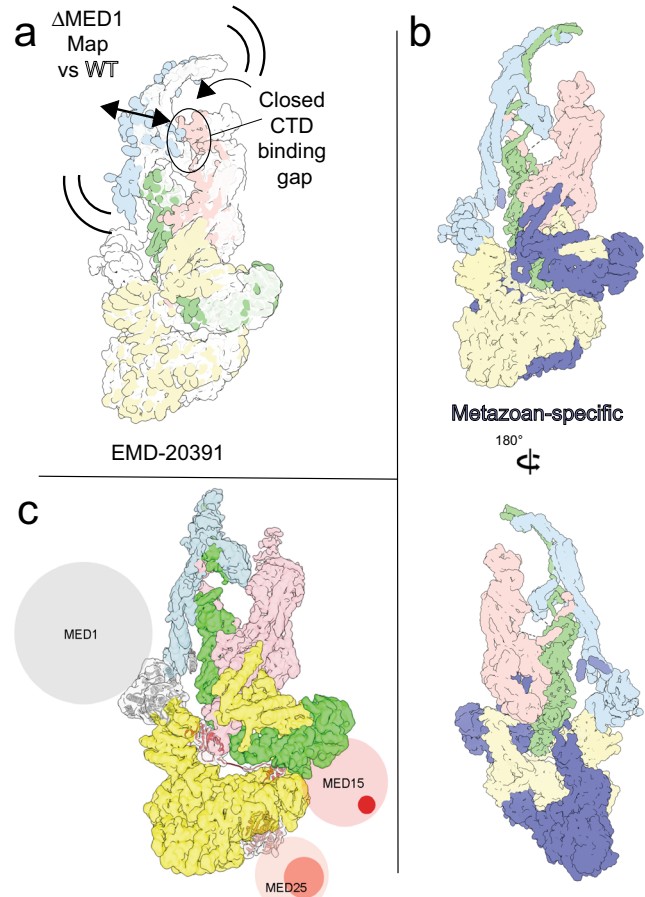

**Fig. 7 Tail and Mediator–RNAPII interactions. a** Cryo-EM analysis of ΔMED1 mMED (EMDB EMD-20391) showed conformational rearrangements of the Middle and Head that closed the CTD-binding gap, consistent with observed RNAPII interaction. **b** Metazoan-specific subunits and subunit domains in mMED contribute to an intricate and very extended Tail–core interaction. **c** Mediator subunits targeted by TFs include large IDRs and at least some of the sites specifically targeted by TFs are included in those IDRs, bringing up the question of how TF binding could trigger Mediator conformational changes.

Ets TF family[29] and HNF4α[13,30] is separated from the N-terminal von Willebrand domain integrated into the Tail by a ~180 aa IDR (Fig. 7c). The presence of IDRs in Mediator subunits that are important activator targets is consistent with involvement of IDRs in hydrophobic interactions that help converge TFs around active genes[31–33]. However, it further complicates the question of TF-induced changes in Mediator conformation. In the case of MED1, MED15, and MED25, Mediator rearrangements would have to result from TF interaction with disordered portions of those subunits, which seems harder to envision. Gene knock out for individual mMED subunits is generally embryonic lethal[34], pointing to an essential role of Mediator in mammalian development and gene expression regulation. However, mMED deletion analysis in cell lines can be used to study the effect of specific subunit deletions. Deletion of MED1 is non-lethal in B, T, or embryonic stem cells and affects expression (>1.5-fold) of ~550 genes (~350 upregulated and ~200 downregulated) in CH12 mouse B cells[14]. Therefore, although MED1 deletion results in mMED conformational changes and considerably increased RNAPII interaction in vitro, it does not lead to general upregulation in vivo. Further studies will be needed to determine how modulation of Mediator conformation and RNAPII interaction

contribute to an overall Mediator mechanism that must involve critical additional aspects.

## Methods

**Purification of mMED for EM studies.** Mouse Mediator was immunopurified from nuclear extracts prepared from CH12 B lymphoma cell lines in which MED19 or MED25 were 3×FLAG-tagged at the N-terminus using CRISPR/Cas9[14]. Additional CRISPR/Cas9 editing was employed to create Mediator subunit truncations in the MED25 FLAG-tagged cell line. Med14 truncation was achieved by dual sgRNA transfection and screening clones for the desired in frame deletion. C-terminal maltose-binding-protein (MBP) tagging of Med15 was performed by overexpression Med15-MBP in Med19 FLAG Med15 knock-out cells[14]. Sequences of sgRNAs and primers for cloning of targeting constructs are listed in Supplementary Table 4. Cell transfection and MBP overexpression was performed as previously described[14]. CH12 cell cultures were expanded to several billion cells in spinner flasks and nuclear extracts were prepared using a published protocol[14]. For mMED purification, nuclear extract was incubated with FLAG M2 agarose resin that had been pre-equilibrated in a buffer containing 50 mM HEPES pH 7.9, 300 mM KOAC, 1 mM EDTA, 10% glycerol, and 0.2% NP-40. After incubation the resin was extensively washed with a buffer containing 50 mM HEPES pH 7.9, 300 mM KOAc, 1 mM EDTA, 10% glycerol, 0.2% NP-40 and 1× mammalian protease inhibitor cocktail (Sigma P8340). This was followed by a second round of washing with a buffer containing 50 mM HEPES pH 7.9, 100 mM KCl, 1 mM EDTA, 5% glycerol and 0.05% NP-40 and no protease inhibitors. Bound mMED was eluted with a buffer containing 500 µg/ml FLAG peptide (Sigma) in 50 mM HEPES pH 7.9, 100 mM KCl, 5% glycerol, 1 mM EDTA and 0.005% NP-40. Purified mMED fractions were flash-frozen in liquid nitrogen and stored at −80 °C until needed for EM studies.

**mMED cryo-EM sample preparation, imaging, and analysis.** MED19-FLAG or MED25-FLAG mMED cryo-EM samples were prepared on lacey carbon grids covered with a thin layer of continuous amorphous carbon (Ted Pella 01824). To prepare cryo-EM samples, purified mMED aliquots were concentrated 20–40 fold using a Vivaspin 500 centrifugal concentrator and 2.5 µL of purified mMED (100–250 µg/mL) were pipetted onto grids plasma-cleaned for 6 s on a Solarus plasma cleaner (Gatan) using an Ar/O₂ gas mixture. Vitrification was performed in liquid ethane using a manual plunge-freeze apparatus. Imaging was performed on Talos Arctica transmission electron microscope (a Thermo Fisher) outfitted with an X-FEG electron source and operating at an acceleration voltage of 200 kV. Automated data collection was carried out using stage-shift targeting in Leginon[35] and four separate image datasets were recorded using a K3 Summit direct electron detector (Gatan). Information about imaging conditions and EM data collection statistics for mMED cryo-EM specimens is summarized in Supplementary Table 2. Cryo-EM movies from both zero-tilt and tilted (20–40°) cryo-EM specimens were recorded to counteract the effect of anisotropic distribution of mMED particle orientations. Image processing was carried out using the CryoSPARC[36] image processing package. Briefly, detector movie frames were subject to patch alignment and patch CTF refinement, followed by automated template-based particle picking. Repeated rounds of 2D image clustering were used to clean the initial image datasets. After this initial cleaning an ab-initio volume was calculated from each dataset and alignment parameters for cryo-EM images were obtained by 3D refinement. Images in each dataset were further screened by 3D image classification and the best images from each dataset were combined. Further rounds of 2D clustering and 3D classification of the combined dataset resulted in a selected set of images that were used for calculation of the final mMED cryo-EM map (Supplementary Figs. 1–3). 2D class averages selected for inclusion of corresponding particles in our cryo-EM analysis of mMED presented in Supplementary Fig. 3b showed no evidence of the presence of either CKM or RNAPII. The amount of CKM and polymerase in our purified Mediator preps was low to begin with and we suspect that the CKM and polymerase tend to dissociate from the small percentage (10–15%) of Mediator particles that initially includes one of them. Although we cannot completely exclude the possibility that a small fraction of mMED particles included in our analysis might have been bound to either the CKM or RNAPII, the 3D classification results presented in Supplementary Fig. 2 provide strong evidence that neither one was present in a meaningful fraction of the particle images included in our analysis and that, if present, they showed high mobility and are unlikely to have had any affect on mMED conformation.

**mMED cryo-EM map interpretation.** Map visualization and interpretation were done using Coot[37] for atomic model building, Phenix[38] for atomic model refinement and Chimera[39] for map visualization. Structure predictions for all mMED subunits were done using Phyre2[40] and I-TASSER[41]. Docking of the *S pombe* Mediator X-ray models derived from cryo-EM and X-ray studies (PDB accession codes 5U0P, 5U0S, and 5N9J), and information about Mediator subunit localization in yeast[9,10] and mammalian[14] Mediators guided initial watershed segmentation (with Chimera) of the mMED cryo-EM map. Atomic model building for mMED subunits was aided by consideration of available yeast Mediator subunit structures and structure prediction results. Atomic model building for Tail model subunits was done de novo for all subunits, except for MED23, for which the X-ray

structure of human MED23 (PDB 6H02) was used as a starting point. Phenix was used for refinement of individual subunit atomic models and for refinement of the overall mMED atomic model (Supplementary Table 2).

**Localization of mMED subunit domains by EM analysis of truncation and MBP-labeled mutants.** Localization of mMED subunit domains through truncation or MBP-tagging was done by EM image analysis of mMED particles preserved in stain. Stained samples were prepared on continuous carbon EM grids (EMS 017543) and preserved using 2% uranyl acetate. Stained samples were imaged using a Talos L120C transmission electron microscope (Thermo-Fisher) outfitted with a LaB₆ filament and operating at an acceleration voltage of 120 kV. Automated data collection was carried out using Leginon[35], with images recorded using a Ceta CMOS detector at a magnification of ×36,000 (corresponding to a pixel size of 3.98 Å). Particles were automatically picked from micrographs with DogPicker[42], using a low threshold to capture all mMED particle images. Roughly 20,000 particle images picked for each experiment were subject to image clustering using ISAC[43]. The cleanest mMED averages were selected and their intensities were normalized to an average density of zero and a standard deviation of 1.0. Multiple average pairs were compared to identify average pairs that would result in the cleanest difference maps. Difference and heat maps were calculated and displayed using a custom image processing and plotting script written in Matlab, which allowed for interactive fine-tuning of cross-correlation-based map alignment prior to difference map calculation. Difference maps were also visualized as heat maps colored by standard deviation, to facilitate their interpretation.

**Reporting summary.** Further information on research design is available in the Nature Research Reporting Summary linked to this article.

## Data availability

Cryo-EM maps and atomic coordinates have been deposited with the Electron Microscopy Data Bank (with accession code EMD-21514) and Protein Data Dank (accession code PDB ID 6W1S). Source data related to Supplementary Fig. 1a are provided with this paper.

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

## Acknowledgements
This work was supported by NIGMS grant R01-67167 (F.J.A.) and by the Intramural Research Programs of NIAMS (R.C.). Cryo-EM data was collected at the Anschutz School of Medicine's Cryo-EM Facility.

## Author contributions
EM experiments were designed by H.Z., N.Y., and F.J.A. H.Z. collected data and calculated the cryo-EM map. N.Y. worked on subunit localization. H.Z. worked on Tail model building. N.Y. worked on core model building and contributed to interpretation and modeling of the Tail. J.K., J.L., and L.E.K. made cell lines and prepared nuclear extracts for Mediator purification. F.J.A. wrote the manuscript, with input from all authors.

## Competing interests
The authors declare no competing interests.
