## [Peer Review File · Nature Communications]

REVIEWER COMMENTS

Reviewer #1 (Remarks to the Author):

In reading the previous reviews and responses it seems that the comments were addressed pretty well. I have some concerns with the current version that will be important to address. The concerns center around inconsistencies in the data and a basic lack of scholarship that undermines the data interpretation.

Major points:

A gel image of the material used for the structural analysis is needed. This was asked for in a prior review. It's a basic minimum requirement so that people can evaluate the quality of the sample.

The MS data in ED figure 1 show the CKM is abundant but it is not seen in the cryo-EM data? The Asturias lab has previously documented the CKM location (Tsai 2013) and it may be too flexible to be resolved here? Some clarification is needed. Also the abundance of RNAP subunits needs to be addressed. Was RNAP holoenzyme seen in the 2D classes? The CKM and RNAP are large and globular such that they should have been identified from the cryo-EM images.

The structure is conserved with yeast but some differences are attributed to conformational changes. Could these conformational changes result from variable association with RNAP or the CKM? Maybe the mammalian Mediator structure is identical to yeast but the contaminants are causing some differences. This should be acknowledged or dismissed based upon new data or added information/explanation.

In the last results section and in the discussion it is stated that MED1 deletion causes "release of the Middle-Tail interaction" but details are not provided. A description such as this implies that the tail dissociates with MED1 deletion but that doesn't seem to be consistent with the data shown or with the models put forward by the authors. This statement should be clarified and re-phrased.

It is stated in the discussion that "deletion of MED1, or disruption of the Tail's integrity, resulted in a marked increase in Mediator-RNAPII interaction" and that it could be explained by a conformation change. The evidence for an increase in RNAP interaction needs to be better justified and this was a concern in the prior reviews also. It is confusing especially because the chart in Figure 7a implies some data to back up the model. The Roeder lab has shown from biochemical experiments that mMED1 "exists predominantly in a TRAP/Mediator subpopulation enriched in RNA polymerase II" based on a title a 2005 Mol Cell paper (Zhang et al.) so how to square with the model put forward here?

The chart in Figure 7a also lists loss of MED23, MED24, MED15, MED16 as promoting RNAP interaction but no evidence is provided and no citation. If you want to show something like this as a main figure you need to have some data to back it up. The chart in 7a implies binding affinities were determined and that doesn't seem to be the case.

In the discussion it is stated that "there is no direct structural evidence for Mediator conformational rearrangements triggered by TF interaction" but this isn't entirely true. The Tijan lab had some old results that made this conclusion. In one study the resolution was poor and the paper is 20 years old (Taatjes et al. Science 2002) but the title notes "activator-induced conformations of the CRSP coactivator".

The discussion mentions recruitment as "almost certain to play a major role in the Mediator mechanism" and this seems to be in response to a reviewer comment. The authors should clarify what is meant by this statement because no one will argue that recruitment is required for a mechanistic role. If it's not there how can it affect RNAP? Making such a statement does not advance

understanding. I think the authors meant to state that Mediator works through RNAP recruitment?

In the discussion it is stated, without citations, that "deletion of MED1 or Tail subunits targeted by TFs does not lead to strong phenotypes." What? MED1 loss is embryonic lethal and MEFs derived from null cells show defects in basic cell functions (Roeder lab). MED24 loss is embryonic lethal (Roeder lab). MED23 loss is embryonic lethal (Berk lab) and so on.

Other comments:

The introduction is confusing because different numbers of subunits are cited for domains, such as 8-9 subunits for Middle or 2-9 subunits for Tail? Which is it? Are the authors implying that different complexes are functioning in cells? Do they see evidence for this in their data?

Typing mistakes on page 4 "225 out of 26" and on page 10 "MmMED"

Reviewer #2 (Remarks to the Author):

The manuscript entitled "Structure of the Mammalian Mediator" submitted by Zhao et al presents an atomic model of the full metazoan mediator complex derived from single particle cryo electron microscopy data. The improved resolution of the map allowed clear description of the tail structure and of the mammalian-specific subunits. The results are clearly described and give an extensive description of the structure showing that yeast and Mouse mediator are mostly conserved, but also describing important new features. Beside the improvement in the structural organization of the Tail module, the manuscript lacks a clear functional message such as interaction with transcription factors, RNA polymerase II interaction or condensate formation, but nevertheless provides new and clearly presented structural data. I would therefore recommend publication in Nature communication provided that the following minor points are addressed.

In the discussion the concept of "recruitment" should be better defined. Do the authors mean, recruitment of mediator to promoter DNA through interactions with TFs? Then the last sentence does not make much sense. "but the fact that deletion of MED1 or Tail subunits targeted by TFs does not lead to strong phenotypes suggests that recruitment might play the dominant role in the Mediator mechanism." Does recruitment mean liquid-liquid phase separation or formation of condensates? The authors claim that the IDPs targeted by the TFs as well as those found in MED1 are instrumental in the formation of LLPS. Here again the last sentence is misleading. The authors should carefully rephrase the text.

Page 2 second § the structure of 25 out of 26 (not 225).

Fig.5e, the residue numbering in the MED15 schematic representation was masked in my version of the figure. The end of the N-terminal ordered region should be indicated in the figure.

Reviewer #3 (Remarks to the Author):

The authors have done a commendable job in revising their manuscript based on the prior reviewers' comments. This work represents an important contribution to Mediator structure and function and I recommend publication in Nature Comm. after addressing the following minor points:

1. At the bottom of pg 3, it's not clear why the concept of phase separation was introduced. None of

the results bear on this point as far as I can tell. I recommend removal.

2. pg 4: "...the structures of 225 out of 26 non-kinase module mMED....." I think that 225 should be 25 – correct?

3. In the cartoon in Fig 1D: Why is Pol shown extended away from Mediator? It makes a close approach in the yeast PIC-Med structure. Also, what is the basis for showing the kinase module attaching only to the hook?

4. Figure S3 is missing the panel designations a,b,c,

5. In the last paragraph of the discussion, there is a discussion of Med's role in recruitment vs conformation-dependent changes in Pol interaction. As written, I'm confused as to what the authors mean by Med's recruitment role. Please clarify in a revised paragraph.

S. Hahn

RESPONSE TO REVIEWER COMMENTS FOR NCOMMS-20-38898-T

Reviewer #1 (Remarks to the Author):

In reading the previous reviews and responses it seems that the comments were addressed pretty well. I have some concerns with the current version that will be important to address. The concerns center around inconsistencies in the data and a basic lack of scholarship that undermines the data interpretation.

Major points:

A gel image of the material used for the structural analysis is needed. This was asked for in a prior review. It's a basic minimum requirement so that people can evaluate the quality of the sample.

We apologize for misunderstanding the feedback provided during the first round of review (“A gel image of the purified material needs to be shown for the readers to have confidence in the quality of the samples. Better yet, mass spectrometry data to give a sense of the possible contaminants present in the samples.”), which we interpreted as requesting mass spectrometry data. We now include SDS-PAGE results in a revised Extended Data Fig 1. We hope that the combination of EM images showing very homogeneous and easily recognizable Mediator particles preserved in stain, MudPIT results and SDS-PAGE analysis will suffice to address the reviewer’s concerns.

The MS data in ED figure 1 show the CKM is abundant but it is not seen in the cryo-EM data? The Asturias lab has previously documented the CKM location (Tsai 2013) and it may be too flexible to be resolved here? Some clarification is needed. Also, the abundance of RNAP subunits needs to be addressed. Was RNAP holoenzyme seen in the 2D classes? The CKM and RNAP are large and globular such that they should have been identified from the cryo-EM images.

The MudPIT results in ED Fig 1 indicate that the CKM and polymerase are present in our purified Mediator preparations but do not provide quantitative information about their relative abundance (e.g., sensitivity is considerably higher for larger proteins that yield more detectable fragments after cleavage). As the reviewer points out, both the CKM and polymerase are large and have a compact and mostly well-folded structure that makes them easy to identify in EM images. This makes it possible to use 2D clustering of EM images of stained specimens to obtain a) a direct and relatively accurate quantification of the amount of CKM and polymerase present in our purified Mediator fractions and b) an initial understanding of CKM and polymerase interaction with mammalian Mediator. We carefully implemented and deployed this EM approach in our published, peer-reviewed studies of wild-type and subunit deletion Mediator interaction with the CKM and polymerase (El Khattabi, et al. 2019. *Cell* 178, 1145–1158 August 22, 2019). We recorded large datasets (>70,000 particles) in stain, exhaustively picked every particle-like feature in all micrographs and ran two independent and comprehensive 2D clustering analyses of the data using two separate programs (Relion and ISAC). Results from these independent duplicate analyses were then used to estimate the amount of CKM and polymerase present in various Mediator preparations by counting images included in each final 2D class (please see Figure RR1 below) and to understand the interaction of both with Mediator. For wild-type (i.e., no subunit deletions), this analysis established that ~75-80% of mMED in our preparations is “free”, 10-15% is bound to the CKM and 10-15% is bound to polymerase (El Khattabi, et al. 2019. *Cell* 178, 1145–1158 August 22, 2019). Therefore, the amounts of CKM and polymerase in our wild-type mMED preparations is roughly a tenth of the amount of free Mediator. Class averages obtained from the stained particle images also showed small amounts of free polymerase and CKM, indicating that they have some tendency to dissociate from Mediator.

In the case of cryo-EM specimens, we routinely see some free CKM and free polymerase averages. However, despite considerable effort, we have never been able to obtain any 2D class averages showing interaction of mMED with either the CKM or polymerase in cryo-EM specimens. Whether this is due to

increased dissociation from mMED upon freezing, or to blurring due to mobility of polymerase and the CKM, we cannot conclusively establish. In stained specimens, variability in the position of the CKM with respect to mMED is much lower than that observed in yeast, which leads us to suspect that dissociation might be a major factor behind the apparent absence of mMED-CKM and (possibly) mMED-polymerase complexes in our cryo-EM specimens.

The structure is conserved with yeast but some differences are attributed to conformational changes. Could these conformational changes result from variable association with RNAP or the CKM? Maybe the mammalian Mediator structure is identical to yeast but the contaminants are causing some differences. This should be acknowledged or dismissed based upon new data or added information/explanation.

Given the rather low sequence identity/homology between corresponding yeast and mammalian Mediator subunits, considerable differences in size between some corresponding subunits, and the presence of several metazoan-specific subunits absent in yeast Mediator, it seems unlikely that the yeast and mammalian Mediators would be “identical” in structure or conformation. In fact, that they are as similar as our studies show them to be seems both interesting and somewhat unexpected.

Because single particle cryo-EM structure determination relies on combining information from many thousands of particle images, the only way to obtain a reasonably detailed map (for mMED 3.5-4.0 Å resolution, at which even bulky amino acid side chains are visible) is to have the overwhelming majority of those images arise from particles whose conformation (and composition) is the same down to the level of detail observed in the final map. This is why, besides the development of direct electron detectors, the biggest contribution to increased resolution cryo-EM maps is the implementation of exhaustive and powerful 2D clustering and 3D classification approaches that make possible identification of particle

images arising from a homogeneous particle subset. 2D class averages selected for inclusion of corresponding particles in our cryo-EM analysis of mMED presented in ED Fig 3B show no indication of the presence of either CKM or polymerase. As explained in response to the previous question, the amount of CKM and polymerase in our purified Mediator preps is low to begin with, and we have evidence that the CKM and polymerase tend to dissociate from the small percentage (10-15%) of Mediator particles that initially includes one of them. The 3D classification results presented in ED Fig 2 provide additional strong evidence that neither polymerase nor the CKM are present in any significant fraction of the particle images included in our analysis.

Again, it is important to consider the meaning of “resolution” in a single particle cryo-EM map. If the overall resolution of the map is 3-5-4.0 Å, this means that the images combined to calculate the map came from particles whose structure is the same to 3.5-4 Å. A map obtained by combining individual particle images is blurred to the maximum resolution of conserved features. Clearly, the differences in conformation between the yeast and mammalian Mediators revealed by our studies reflect changes many times larger than 4 Å. These considerations will be obvious to readers that understand how cryo-EM works and look at our Extended Data.

In the last results section and in the discussion it is stated that MED1 deletion causes “release of the Middle-Tail interaction” but details are not provided. A description such as this implies that the tail dissociates with MED1 deletion but that doesn’t seem to be consistent with the data shown or with the models put forward by the authors. This statement should be clarified and re-phrased.

Thank you for pointing out that the meaning of that statement was not clear. We have edited the text to clarify what we meant: the only interaction between the Middle and Tail modules involves the structured N-terminal portion of MED1 (Middle) and the N-terminal portion of MED24 (Tail). Therefore, deletion of MED1 eliminates Middle-Tail interaction. We have made this point in both Results (p. 10) and Discussion (p. 11)

It is stated in the discussion that “deletion of MED1, or disruption of the Tail’s integrity, resulted in a marked increase in Mediator-RNAPII interaction” and that it could be explained by a conformation change. The evidence for an increase in RNAP interaction needs to be better justified and this was a concern in the prior reviews also. It is confusing especially because the chart in Figure 7a implies some data to back up the model. The Roeder lab has shown from biochemical experiments that mMED1 “exists predominantly in a TRAP/Mediator subpopulation enriched in RNA polymerase II” based on a title a 2005 Mol Cell paper (Zhang et al.) so how to square with the model put forward here?

We find this comment confusing. A reference is clearly included right in the middle of the relevant sentence in the Discussion (p. 11):

“We previously reported results from EM analysis of mMED subunit deletion mutants ¹⁴ showing that release of core-Tail interactions by deletion of MED1, or disruption of the Tail’s integrity, resulted in a marked increase in Mediator-RNAPII interaction (Fig. 7a). In the case of MED1 deletion, we were able to show that the increase in RNAPII interaction might be explained by a change in the conformation of mMED.”

The relevant peer-reviewed results (from El Khattabi, et al. 2019. *Cell* 178, 1145–1158 August 22, 2019) are shown here in **Figure RR1** above. The next panel in the same figure of that Cell paper shows how deletion of MED1 causes a rearrangement of the CTD-binding gap between the knob and neck domains.

We are familiar with the report from the Roeder lab suggesting a link between MED1 and Pol II interaction with Mediator. As evidenced in 2D class averages from both stained and cryo specimens, in results from 3D classification and in the final 3D cryo-EM map, we see no indication of MED1 dissociation in our purified mMED preparations, except when the subunit is explicitly deleted from Mediator. We cannot speculate on the reasons why the report from the Roeder lab might have detected a less than stoichiometric amount of MED1 and linked that to polymerase interaction.

The chart in Figure 7a also lists loss of MED23, MED24, MED15, MED16 as promoting RNAP interaction but no evidence is provided and no citation. If you want to show something like this as a main figure you need to have some data to back it up. The chart in 7a implies binding affinities were determined and that doesn't seem to be the case.

A citation to peer-reviewed results was included in the original text, as we indicated in response to the previous question. We have now also added that citation to the figure legend to further clarify the source of the information in Fig. 7a. The peer-reviewed results on which the diagram in Fig. 7a are based are shown here in Fig. RR1 (p. 2 of this reply) and the methodology to obtain those results was detailed in response to an earlier question. Hopefully this clarifies how we were able to obtain evidence for increased polymerase interaction with mMED subunit deletion mutants. Someone reading our report will be able to go to the referenced publication and see the results on which the diagram is based. Our goal is to have Fig. 7 work as a "discussion" figure that presents a visual summary of our conclusions and we purposefully presented the information in Fig 7a in a way that hopefully highlights that we have observed changes in polymerase interaction with Mediator, but that avoids the impression that we are making precise quantitative claims.

In the discussion it is stated that "there is no direct structural evidence for Mediator conformational rearrangements triggered by TF interaction" but this isn't entirely true. The Tjian lab had some old results that made this conclusion. In one study the resolution was poor and the paper is 20 years old (Taatjes et al. Science 2002) but the title notes "activator-induced conformations of the CRSP coactivator".

We are very familiar with those seminal papers from the Tjian lab and have cited them in our own publications. Now that we have a model of mammalian Mediator derived from a 3.5-4.0 Å cryo-EM map, it has become clear that the limitations of older EM studies go beyond limited (<30 Å) resolution. Beyond rough similarity in overall shape, there is little in common between those old structures and the current ones. Some years ago, we made extensive attempts to match the Tjian structures to that of the *S cerevisiae* Mediator and, beyond being able to correctly distinguish the end of the old maps corresponding to the Tail module, we could not match them in any way. Purported location of "binding sites" does not match the location of target subunits (e.g. the binding site for the CTD was "localized" to what is now known to be the Tail module) and now that we understand the Mediator structure and the rearrangements that the various modules can undergo, it has become clear that a) the "activator-induced" conformations do not resemble any conformation of Mediator ever seen in more recent EM studies, b) the structural rearrangements that would be required to reach the purported "activator-bound" conformations seem entirely incompatible with what we now know Mediator modules would be able to accommodate.

We continue to believe that activators might be able to induce Mediator conformational rearrangements, but now realize that the original papers from the Tjian group might have been correct in spirit, but not factually correct. This is understandable, as dataset included comparatively very few particles 20 years ago and 2D clustering and 3D classification algorithms were considerably less powerful and, if applied, would have been severely limited by the small number of images. This was a common problem and our own studies suffered from the same limitation. For all of these reasons, we feel that it is no longer appropriate to cite the original Tjian studies as a reliable source of information about activator-induced conformational changes. To acknowledge the existence of the pioneering Tjian publications we have

revised the sentence cited by the reviewer to read “there is no conclusive structural evidence for Mediator conformational rearrangements triggered by TF interaction” (p. 12).

The discussion mentions recruitment as “almost certain to play a major role in the Mediator mechanism” and this seems to be in response to a reviewer comment. The authors should clarify what is meant by this statement because no one will argue that recruitment is required for a mechanistic role. If it’s not there how can it affect RNAP? Making such a statement does not advance understanding. I think the authors meant to state that Mediator works through RNAP recruitment?

We greatly appreciate the feedback from the reviewers, which prompted us to discuss in more detail, and hopefully more clearly, the implications of our results. We have re-written the Discussion to put the results described in our manuscript (i.e., mMED structure and a connection between mMED core rearrangements and mMED Pol II interaction) in context and have eliminated speculation about additional aspects of the Mediator mechanism that were not addressed by our studies.

In the discussion it is stated, without citations, that “deletion of MED1 or Tail subunits targeted by TFs does not lead to strong phenotypes.” What? MED1 loss is embryonic lethal and MEFs derived from null cells show defects in basic cell functions (Roeder lab). MED24 loss is embryonic lethal (Roeder lab). MED23 loss is embryonic lethal (Berk lab) and so on.

We completely agree that this statement was confusing. In a re-written Discussion we now clarify this point and include appropriate references:

“Gene knock out for individual mMED subunits is generally embryonic lethal³⁴, pointing to an essential role of Mediator in mammalian development and gene expression regulation. However, mMED deletion analysis in cell lines can be used to study the effect of specific subunit deletions. Deletion of MED1 is non-lethal in B, T or embryonic stem cells and affects expression (>1.5-fold) of ~550 genes (~350 upregulated and ~200 downregulated) in CH12 mouse B cells¹⁴.”

Other comments:

The introduction is confusing because different numbers of subunits are cited for domains, such as 8-9 subunits for Middle or 2-9 subunits for Tail? Which is it? Are the authors implying that different complexes are functioning in cells? Do they see evidence for this in their data?

We apologize for the confusion and have edited the text to clarify that the specific number of subunits in each module varies among different eukaryotic organisms.

Typing mistakes on page 4 “225 out of 26” and on page 10 “MmMED”

We apologize for overlooking that mistake, which has been fixed.

Reviewer #2 (Remarks to the Author):

The manuscript entitled “Structure of the Mammalian Mediator” submitted by Zhao et al presents an atomic model of the full metazoan mediator complex derived from single particle cryo electron microscopy data. The improved resolution of the map allowed clear description of the tail structure and of the mammalian-specific subunits. The results are clearly described and give an extensive description of the structure showing that yeast and Mouse mediator are mostly conserved, but also describing important new features. Beside the improvement in the structural organization of the Tail module, the manuscript lacks a clear functional message such as interaction with transcription factors, RNA polymerase II interaction or condensate formation, but nevertheless provides new and clearly presented structural

data. I would therefore recommend publication in Nature communication provided that the following minor points are addressed.

In the discussion the concept of “recruitment” should be better defined. Do the authors mean, recruitment of mediator to promoter DNA through interactions with TFs? Then the last sentence does not make much sense. “but the fact that deletion of MED1 or Tail subunits targeted by TFs does not lead to strong phenotypes suggests that recruitment might play the dominant role in the Mediator mechanism.” Does recruitment mean liquid-liquid phase separation or formation of condensates? The authors claim that the IDPs targeted by the TFs as well as those found in MED1 are instrumental in the formation of LLPS. Here again the last sentence is misleading. The authors should carefully rephrase the text.

We greatly appreciate the feedback from the reviewers, which prompted us to discuss in more detail, and hopefully more clearly, the implications of our results. We have re-written the Discussion to put the results described in our manuscript (i.e., mMED structure and a connection between mMED core rearrangements and mMED Pol II interaction) in context and have eliminated speculation about additional aspects of the Mediator mechanism that were not addressed by our studies.

Page 2 second § the structure of 25 out of 26 (not 225).

Thank you for pointing this out. We apologize for the typo, which has now been corrected to indicate “25 out of 26 non-kinase module subunits”

Fig.5e, the residue numbering in the MED15 schematic representation was masked in my version of the figure. The end of the N-terminal ordered region should be indicated in the figure. We have fixed those problems and also rearranged Fig 5e to make it easier to correlate the schematic representation of MED15 to the model we derived from the cryo-EM map.

Reviewer #3 (Remarks to the Author):

The authors have done a commendable job in revising their manuscript based on the prior reviewers’ comments. This work represents an important contribution to Mediator structure and function and I recommend publication in Nature Comm. after addressing the following minor points:

1. At the bottom of pg 3, it’s not clear why the concept of phase separation was introduced. None of the results bear on this point as far as I can tell. I recommend removal.

This sentence was left over from the previous version of the paper that included some results related to phase separation. We completely agree with the reviewer’s comment and have removed that sentence.

2. pg 4: “...the structures of 225 out of 26 non-kinase module mMED.....” I think that 225 should be 25 – correct?

Thank you for pointing this out. We apologize for the typo, which has now been corrected to indicate “25 out of 26 non-kinase module subunits”

3. In the cartoon in Fig 1D: Why is Pol shown extended away from Mediator? It makes a close approach in the yeast PIC-Med structure. Also, what is the basis for showing the kinase module attaching only to the hook?

Our goal was to have the cartoon in Fig 1D reflect as closely as possible the EM data we have for mammalian Mediator. We do not have any cryo-EM data (neither 2D class average nor 3D maps) showing interaction of mammalian Mediator with Pol II or with the kinase module. The only information we have

about those interactions comes from 2D class averages calculated from particles preserved in stain. In those 2D class averages from stained data Pol II appears at a number of different positions, mostly around the hook and rarely at any position consistent with the position of polymerase in the yeast holoenzyme. This suggests that our images of stained particles only show polymerase tethered to mammalian Mediator through the CTD and not forming a true holoenzyme complex like the one observed in yeast. Since we do not have any EM data consistent with the presence of a full mammalian holoenzyme complex in our samples or providing any information about the structure of that hypothetical mammalian holoenzyme, the cartoon in Fig 1D shows what we have observed: polymerase tethered to Mediator through the CTD. Similar reasoning explains the depiction of the kinase module interaction with mammalian Mediator. 2D class averages from stained specimens clearly show the kinase module interacting with the hook as shown in Fig 1D but show no other Mediator-kinase module contacts. As with polymerase, we felt that the cartoon in Fig 1D should reflect what we have observed in our EM studies.

4. Figure S3 is missing the panel designations a,b,c,

We apologize for that mistake. The panel labels were accidentally left out during conversion of the file to PNG format and are now visible in a revised figure.

5. In the last paragraph of the discussion, there is a discussion of Med's role in recruitment vs conformation-dependent changes in Pol interaction. As written, I'm confused as to what the authors mean by Med's recruitment role. Please clarify in a revised paragraph.

We greatly appreciate the feedback from the reviewers, which prompted us to discuss in more detail, and hopefully more clearly, the implications of our results. We have re-written the Discussion to put the results described in our manuscript (i.e., mMED structure and a connection between mMED core rearrangements and mMED Pol II interaction) in context and have eliminated speculation about additional aspects of the Mediator mechanism that were not addressed by our studies.

REVIEWERS' COMMENTS

Reviewer #1 (Remarks to the Author):

I apologize if the authors mis-interpreted my previous comments as too obstructive. That was not my intent and I hope to make that more clear below. I think we may agree to disagree on some points. The new structural data are excellent and that is the basis for consideration in a high-impact journal such as Nature Communications. I have additional remarks for the authors to consider, but I will leave it to them whether to make any changes.

I think the authors should reconsider Figure 7A. The conclusions are based on 2D clustering from negative stain data. The samples were generated from a Med19 IP in which CKM and polymerase were found to co-purify. The authors acknowledge that the CKM and polymerase can dissociate during EM sample preparation because free CKM and free polymerase can be imaged. Can't you see the problems here? There are many: accuracy of 2D projections to call associated polymerase vs. CKM vs. something else, low resolution and high noise of 2D projections, use of such data to infer binding affinities (still implied from 7A), sample-to-sample or extract-to-extract variability in co-purifying CKM or polymerase, no reporting on statistical significance among comparisons, etc.

I am trying to prevent over-interpretation of these types of data, for two reasons. One, if published in a top journal by one of the top structural biologists in the world, people will believe it to be true, but in fact the data are not reliable or conclusive. Two, as a top structural biologist (Asturias), this sets an example and an expectation that others can use similar types of data to draw concrete conclusions about protein-protein interactions. Elsewhere in the response the authors describe in great detail their reservations about other works because of low resolution. Please try to be consistent. It is also frustrating that the authors are insistent to include 7A, since these data from 2019 are not needed and distract from the new and more important/more reliable results shown in figures 1-6. To be blunt, I think the authors undermine their credibility by including figure 7A.

A remaining request for the authors to consider is related to previous comments about RNAP and CKM co-purification with Mediator. The authors provide an excellent and insightful description of why this does not negatively impact the data analysis, and conclude by stating that these detailed considerations "will be obvious to readers that understand how cryo-EM works" and that's true. The point I'm trying to make is that it doesn't matter whether the reviewers or the authors understand this, because once published, it will be most important that readers understand this. Nature Communications is not a specialized structural biology journal. It has a broad and general audience and the authors would benefit the community by explaining how potential issues with heterogeneity can be mitigated, perhaps as part of the legend in ED figure 2. This will also benefit the authors because readers will be able to appreciate the extensive and careful work that resulted in the high-quality structures shown here. This was also a basis for the request to show the gel image, because that is something readily available and understandable to a broad audience.

Reviewer #2 (Remarks to the Author):

The authors addressed my questions correctly. I would recommend publication of the manuscript.

Reviewer #3 (Remarks to the Author):

The authors have done a nice job in addressing the reviewers' comments and in revising the manuscript. I recommend publication.

One minor point to be addressed:

Extended data, Fig 1:

The authors have added an image of a stained protein gel of the Mediator prep used for the cryoEM studies in response to another reviewer. However, there are a number of samples between the size std and the Med prep in the gel that are not labeled. Please update the figure and legend, or remove the irrelevant gel lanes from the figure. Also, if the authors know the identity of the prominent protein band in the Med prep (~ 45 kDa), please identify.

RESPONSE TO REVIEWERS' COMMENTS

Responses in RED.

Reviewer #1 (Remarks to the Author):

I apologize if the authors mis-interpreted my previous comments as too obstructive. That was not my intent and I hope to make that more clear below. I think we may agree to disagree on some points. The new structural data are excellent and that is the basis for consideration in a high-impact journal such as Nature Communications. I have additional remarks for the authors to consider, but I will leave it to them whether to make any changes.

I think the authors should reconsider Figure 7A. The conclusions are based on 2D clustering from negative stain data. The samples were generated from a Med19 IP in which CKM and polymerase were found to co-purify. The authors acknowledge that the CKM and polymerase can dissociate during EM sample preparation because free CKM and free polymerase can be imaged. Can't you see the problems here? There are many: accuracy of 2D projections to call associated polymerase vs. CKM vs. something else, low resolution and high noise of 2D projections, use of such data to infer binding affinities (still implied from 7A), sample-to-sample or extract-to-extract variability in co-purifying CKM or polymerase, no reporting on statistical significance among comparisons, etc. I am trying to prevent over-interpretation of these types of data, for two reasons. One, if published in a top journal by one of the top structural biologists in the world, people will believe it to be true, but in fact the data are not reliable or conclusive. Two, as a top structural biologist (Asturias), this sets an example and an expectation that others can use similar types of data to draw concrete conclusions about protein-protein interactions. Elsewhere in the response the authors describe in great detail their reservations about other works because of low resolution. Please try to be consistent. It is also frustrating that the authors are insistent to include 7A, since these data from 2019 are not needed and distract from the new and more important/more reliable results shown in figures 1-6. To be blunt, I think the authors undermine their credibility by including figure 7A.

We appreciate the Reviewer's advice and have removed the original panel 7a in a new revised version of Figure 7.

A remaining request for the authors to consider is related to previous comments about RNAP and CKM co-purification with Mediator. The authors provide an

excellent and insightful description of why this does not negatively impact the data analysis, and conclude by stating that these detailed considerations “will be obvious to readers that understand how cryo-EM works” and that’s true. The point I’m trying to make is that it doesn’t matter whether the reviewers or the authors understand this, because once published, it will be most important that readers understand this. Nature Communications is not a specialized structural biology journal. It has a broad and general audience and the authors would benefit the community by explaining how potential issues with heterogeneity can be mitigated, perhaps as part of the legend in ED figure 2. This will also benefit the authors because readers will be able to appreciate the extensive and careful work that resulted in the high-quality structures shown here. This was also a basis for the request to show the gel image, because that is something readily available and understandable to a broad audience.

We appreciate the Reviewer’s concern and have added the following paragraph in the “**mMED cryo-EM sample preparation, imaging and analysis.**” subsection in METHODS:

“2D class averages selected for inclusion of corresponding particles in our cryo-EM analysis of mMED presented in Supplementary Figure 3b showed no evidence of the presence of either CKM or RNAPII. The amount of CKM and polymerase in our purified Mediator preps was low to begin with and we suspect that the CKM and polymerase tend to dissociate from the small percentage (10-15%) of Mediator particles that initially includes one of them. Although we cannot completely exclude the possibility that a small fraction of mMED particles included in our analysis might have been bound to either the CKM or RNAPII, the 3D classification results presented in Supplementary Figure 2 provide strong evidence that neither one was present in a meaningful fraction of the particle images included in our analysis and that, if present, they showed high mobility and are unlikely to have had any affect on mMED conformation.”

Reviewer #2 (Remarks to the Author):

The authors addressed my questions correctly. I would recommend publication of the manuscript.

We sincerely appreciate the feedback from the reviewer, which helped improve our manuscript.

Reviewer #3 (Remarks to the Author):

The authors have done a nice job in addressing the reviewers’ comments and in revising the manuscript. I recommend publication.

One minor point to be addressed:

Extended data, Fig 1:

The authors have added an image of a stained protein gel of the Mediator prep used for the cryoEM studies in response to another reviewer. However, there are a number of samples between the size std and the Med prep in the gel that are not labeled. Please update the figure and legend, or remove the irrelevant gel lanes from the figure. Also, if the authors know the identity of the prominent protein band in the Med prep (~ 45 kDa), please identify.

We have edited an image of the original gel to keep only the lane with MW markers and the lane for the purified mMED fraction used for cryo-EM sample preparation. In editing the image of the gel, we followed the instructions from the editors: the two lanes included in the new image are separated by a black line and an image of the original gel has been uploaded as a Source Data file.

We suspect that the band at ~45kDa corresponds to actin, which is the major contaminant identified by MudPIT analysis. We did not label the band in the gel because we did not perform mass spectrometry analysis to confirm our tentative identification.